# New Insights into Development of Transglutaminase 2 Inhibitors as Pharmaceutical Lead Compounds

**DOI:** 10.3390/medsci6040087

**Published:** 2018-10-08

**Authors:** Soo-Youl Kim

**Affiliations:** Tumor Microenvironment Research Branch, Division of Cancer Biology, Research Institute, National Cancer Center, Goyang 10408, Korea; kimsooyoul@gmail.com

**Keywords:** transglutaminase 2, inhibitor, conformation, dimer, polymerization, cross-linking

## Abstract

Transglutaminase 2 (EC 2.3.2.13; TG2 or TGase 2) plays important roles in the pathogenesis of many diseases, including cancers, neurodegeneration, and inflammatory disorders. Under normal conditions, however, mice lacking TGase 2 exhibit no obvious abnormal phenotype. TGase 2 expression is induced by chemical, physical, and viral stresses through tissue-protective signaling pathways. After stress dissipates, expression is normalized by feedback mechanisms. Dysregulation of TGase 2 expression under pathologic conditions, however, can potentiate pathogenesis and aggravate disease severity. Consistent with this, TGase 2 knockout mice exhibit reversal of disease phenotypes in neurodegenerative and chronic inflammatory disease models. Accordingly, TGase 2 is considered to be a potential therapeutic target. Based on structure–activity relationship assays performed over the past few decades, TGase 2 inhibitors have been developed that target the enzyme’s active site, but clinically applicable inhibitors are not yet available. The recently described the small molecule GK921, which lacks a group that can react with the active site of TGase 2, and efficiently inhibits the enzyme’s activity. Mechanistic studies revealed that GK921 binds at an allosteric binding site in the N-terminus of TGase 2 (amino acids (a.a.) 81–116), triggering a conformational change that inactivates the enzyme. Because the binding site of GK921 overlaps with the p53-binding site of TGase 2, the drug induces apoptosis in renal cell carcinoma by stabilizing p53. In this review, we discuss the possibility of developing TGase 2 inhibitors that target the allosteric binding site of TGase 2.

## 1. Introduction

### 1.1. Prospective Benefits of Therapeutic Approach through TGase 2 Inhibition In Clinical Side

Physiological functions and reaction mechanisms of transglutaminase 2 (TGase 2) have been studied for over six decades. Transglutaminase 2 is widely distributed in tissues, and is proposed to have various functions in a context-dependent fashion [1,2,3]. Transglutaminase 2 is responsible for the change of normal physiology including fibroblast function [4], wound healing [5], clearance of apoptotic cells [6], macrophage phagocytosis [7], glucose tolerance [8] as well as playing a key role in the development of disease pathogenesis including various cancers [9,10,11], and several neurological disorders including Huntington’s, Alzheimer’s and Parkinson’s [12,13,14]. More reactions and biological functions of TGase 2 are discussed in the reviews [1,15] including nuclear factor κB (NF-κB) activation through NF-κB inhibitor α (I-κBα) inhibition [16], hypoxia-inducible factor 1α (HIF-1α) activation through von Hippel–Lindau tumor suppressor (VHL) inhibition [17] and suppression of apoptosis in cancer through p53 inhibition [15,18]. 

It seems that irreversible inhibitors are attractive for the development of TGase 2 inhibitors. In the field of TGase 2 inhibitor development as a prospective clinical candidate, most research endeavors focus on targeting the active site as an irreversible inhibitor [19]. Several clinical trials are being tested for TGase 2 inhibitors. Cysteamine is known to target active site cysteine residues and has been launched as a potential therapeutic for a broad range of disease indications including cystic fibrosis [20], neurodegenerative diseases [21], and Huntington’s disease [22], amongst others by Raptor pharmaceuticals, Mylan, and the European Institute for Cystic Fibrosis Research [19]. Zedira has focused on developing peptidomimetic TGase 2 inhibitors which mimick the TGase 2 substrate for the treatment of celiac disease. The Cure Huntington’s Disease Initiative Foundation (CHDI) and Evotec are developing a different class of Michael acceptor compounds targeting the TGase 2 active site for the treatment of diseases such as neurodegenerative diseases and celiac disease [23]. Although irreversible TGase 2 inhibitors are dominant in the development pipeline, a new finding of the GK921 mechanism in TGase 2 inhibition [24] suggests that reversible inhibitors should be considered as important potentials for pharmaceutical lead compounds. 

### 1.2. Is the Active Site of TGase 2 the Unique Target for Inhibition of Enzyme Activity?

The transglutaminase 2 (TGase 2, E.C. 2.1.2.13) inhibitor GK921, 3-(phenylethynyl)-2-(2-(pyridin-2-yl)ethoxy)pyrido[2,3-*b*]pyrazine, eliminates renal cell cancer (RCC) in a xenograft model by inhibiting TGase 2 [25]. However, it does not contain a ‘warhead’ that attacks the active site of the enzyme [19]. Indeed, GK921 does not interact with the active site of TGase 2 at all, but its binding to the N-terminus (amino acids (a.a.) 81–116) inactivates TGase 2 by inducing a conformational change that accelerates the noncovalent self polymerization of the protein [24] (Figure 1). Because the GK921-binding site overlaps with that of p53, the compound also prevents TGase 2 from binding to p53 in RCC cells, thereby stabilizing the tumor suppressor protein [24] (Figure 1). Inhibition of TGase 2 activity through binding at the allosteric site, in conjunction with stabilization of p53, yields significant anticancer effects in a RCC model [25]. This suggests that TGase 2 inhibitors could be developed that target the allosteric site; although many efforts showed that development of TGase 2 inhibitor was progressed through the structure activity relationship (SAR) analysis targeting active site [23,26,27].

### 1.3. Is TGase 2 active as a Monomer?

To distinguish it from the blood clotting Factor XIIIa (FXIIIa) [33], TGase 2 purified in its active form from cytosolic fractions of various tissues was named tissue transglutaminase [34] or transglutaminase C (‘C’ for cytosol) [35,36]. Blood clotting Factor XIIIa is synthesized as a proenzyme that is activated by thrombin, and subsequently forms a tetramer with a homodimer of inactive FXIIIb or a homodimer of FXIIIa [37]. By contrast, TGase 2 is considered to be active as a monomer because it is purified in monomeric form [38]. However, TGase 2 is inactive until it binds to calcium, and X-ray crystallography has shown that it can adopt a dimeric form [39] (Figure 2B). When TGase 2 is purified at 4 °C without guanosine triphosphate (GTP), it exists mostly in the unfolded monomeric form [39]. However, the enzyme changes its conformation in a temperature-dependent manner, and largely forms dimers at temperatures above 30 °C [29]. Consequently, TGase 2 is an unfolded inactive dimer at 37 °C [29]. The homodimer-binding domain of TGase 2 is at the C-terminus (a.a. 593–600). Thus, like FXIIIa, TGase 2 can exist in an inactive dimer form under certain conditions.

### 1.4. Does TGase 2 Change Conformation by Activation?

Crystallographic analysis of TGase 2 bound to the inhibitor Ac-P(6-diazo-5-oxo-norleucine)LPF-NH_2_ revealed an unfolded stretched form that is thought to represent the active state; the inhibitor sequence was derived from TGase 2 substrate in gluten, PQPQLPY [40] (Figure 2B). This observation suggests that TGase 2 changes its conformation from a folded structure bound to GTP into a stretched, unfolded active state [39]. However, the unfolded form is still inactive in the absence of bound calcium. The conformational change that occurs in unfolded TGase 2 upon calcium binding remains obscure because, once activated by calcium, the enzyme cross-links its substrates, including itself, very quickly [41].

An extracellular matrix (ECM) study revealed that TGase 2 can form a triple complex with fibronectin and integrin β1 that accelerates cell adhesion and proliferation in cancer cells [42]. Interestingly, in RCC, TGase 2 forms another triple complex, in this case with p53 and p62, resulting in depletion of p53 from the autophagosome [30,43]. In both cases, TGase 2 is thought to simultaneously bind in the unfolded state to two different molecules.

At 37 °C, TGase 2 exists either as a folded monomer bound to GTP or as an unfolded homodimer, either of which can also form complexes with other binding proteins: “substrate1–TGase 2–TGase 2–substrate2” or “substrate1–TGase 2–substrate2,” respectively. Ultimately, the enzyme is activated by calcium binding, and the active form performs the cross-linking reaction (Figure 2C). Thus, TGase 2 activation absolutely requires calcium and conformational change. Consequently, the TGase 2 inhibitors that were developed based on structure–activity relationship (SAR) studies, which target the folded monomeric form of TGase 2, may not be effective under physiological conditions.

## 2. Dynamics of TGase 2 Conformation and Activity

For many years after the first crystal structures of TGase 2 were published, the enzyme was thought to exist exclusively in the folded monomeric form [39,44]. However, a subsequent crystal structure of TGase 2 in complex with a substrate mimic revealed that a new, unfolded conformation of the enzyme represented the true active form [40]. This finding suggests that TGase 2 must undergo a large conformational change, akin to turning itself inside out, during catalysis.

### 2.1. Proposal: The TGase 2 Dimer Is the Active Enzyme

Given that its acyl-donor and acyl-acceptor pockets are so small [40,45], it remains unknown how TGase 2 catalyzes cross-linking between two huge proteins, e.g., ankyrin-1 (~200 kDa), myosin (~200 kDa), filamin-A (~300 kDa), and so on (for a list of known TGase 2–binding proteins, see [46]). To ensure successful cross-linking, TGase 2 must hold both substrates simultaneously and push them into the small reactive pockets at the same time, raising the question of whether monomeric TGase 2 is capable of cross-linking two larger proteins. Blood clotting Factor XIIIa provides a hint regarding how the TGase 2-catalyzed reaction generates intermolecular cross-links. Blood clotting Factor XIIIa (A-enzyme) exists as a heterotetramer (A_2_B_2_) with FXIIIb (B-binding subunit) that is activated by thrombin and induces binding of fibrin to form a clot. Activated FXIII produces cross-links between γ-chains of two neighboring fibrin molecules in the longitudinal orientation of the clot [47,48]. Cross-linking occurs between Gln398 or 399 on the γ-chain of one fibrin molecule and Lys406 on the γ-chain of another, resulting in the formation of two antiparallel isopeptide bonds that connect the D-regions of both fibrinogen molecules [47,49]. Thus, to achieve intermolecular cross-links between fibrinogen molecules, FXIIIa must be a homodimer (~340 kDa). To use this reaction mechanism, TGase 2 must also form a dimer.

Observations made in studies of celiac disease provide insight into whether TGase 2 could indeed form such a complex. In celiac disease, TGase 2 has been identified as a component of a neoantigen formed by cross-linking of the enzyme to gluten [50]. Transglutaminase 2–specific B cells efficiently recognize a multimeric form of TGase 2; due to the conformation-dependent binding of the C-terminal domain, TGase 2 itself can serve as a substrate for cross-linking [51]. The authors of that study proposed the existence of an ionically bonded complex, gluten–TGase 2–TGase 2–gluten, that is converted into a covalently cross-linked complex after the interaction with calcium [51]. In hepatocytes, TGase 2 also contributes to the formation of ECM, a process in which TGase 2 itself is incorporated into a high-molecular weight extracellular protein complex (~200 kDa) [52]. Mass spectroscopy revealed that a large insoluble polymer (<300 kDa) detected in cancer cells contains eukaryotic translation initiation factor-1α (eIF-1α), annexin A1, fibronectin, tubulins, ribonucleoproteins, TGase 2 itself, and other molecules [53]. The same type of high-molecular weight polymer was also observed after incubation of purified guinea pig liver transglutaminase in vitro [41]. Therefore, TGase 2 can form homodimers and heteropolymers via noncovalent binding under physiological conditions, and later become covalently cross-linked into a very large protein mass following activation by calcium.

### 2.2. TGase 2 Dimer Formation and Activity

Enzyme folding is a spontaneous process that is mainly guided by hydrophobic interactions, formation of intramolecular hydrogen bonds, and van der Waals forces, and opposed by conformational entropy [54]. Minimization of the number of hydrophobic side chains exposed to water is an important driving force behind the folding process [54]. Based on this fundamental principle of biochemical thermodynamics, the conformational change of TGase 2 from the folded structure to the unfolded stretched structure, due to binding of either inhibitor or substrate, is potentially dangerous to the cell because the enzyme is at high risk of aggregation due to exposure of its hydrophobic region. Therefore, an unfolded dimer of TGase 2 or a triple complex of unfolded TGase 2 with other proteins is likely to be more stable than the unfolded monomer alone. Molecular modeling using information from TGase 2 revealed that FXIIIa can form a dimer in the unfolded state [37]. However, the idea is not consistent with previous reports showing that TGase 2 yields a single peak in chromatographic purification and a single band in gel electrophoresis [34].

Recent observations may be able to resolve this discrepancy. We found that TGase 2 switches its conformation from a monomer to a dimer following a change in temperature [29] (Figure 2C). In particular, the conformation switched to an unfolded dimer above 30 °C, but the protein remains as an unfolded monomer below 30 °C [29]. In the presence of calcium below 30 °C, monomer TGase 2 catalyzes an intramolecular cross-link [29]. By contrast, in the presence of calcium above 30 °C, dimeric TGase 2 catalyzes an intermolecular cross-link [29]. Thus, TGase 2 forms a single peak in chromatography at 4 °C because it tends to be monomeric below 30 °C. The C-terminus of TGase 2 (a.a. 593–600) is the critical region for dimerization [51], and this region also contains the phosphoinositide-binding site (a.a. 590–602) [55]. The unconventional secretion of TGase 2 described in the next section involves phospholipid-dependent delivery of the protein into recycling endosomes, based on the protein’s ability to bind to phosphoinositides on endosomal membranes [55].

### 2.3. Multiple Complexes of TGase 2 in the ECM

Transglutaminase 2, which does not have a secretory signal peptide, is secreted into the ECM through unconventional pathways, including microvesicles [56]. In the ECM, TGase 2 is considered to act both as a structural protein and a signaling modifier, playing important enzymatic and nonenzymatic roles [57]. Enzymatically, TGase 2 cross-links ECM proteins to stabilize the matrix overall. Nonenzymatically, TGase 2 modulates interactions of cells with the ECM by strongly associating with integrins, syndecan-4, and growth factor receptors [57]. This nonenzymatic binding of TGase 2 induces strong cell–cell adhesion through direct noncovalent interactions involving the unfolded conformation (homodimers, heterodimers, heterotrimers, heterotetramers, etc.) [57]. For example, TGase 2 promotes cell adhesion by associating with fibronectin via the gelatin-binding region, which does not overlap with the major integrin-binding sites [58]. At the same time, TGase 2 is also strongly associated with integrins, and promotes the interactions of cells with fibronectin by forming a bridge with integrins [57]. Theoretically, the triple heterocomplex fibronectin–TGase 2–integrin may be a major complex in the ECM. However, TGase 2 may itself form a homodimer that can convert the triple complex into multiple heterocomplexes, thereby promoting cell adhesion in specific areas.

Recent studies of renal fibrosis revealed that TGase 2 contains a sequence important for extracellular trafficking at the N-terminus (a.a. 88–106) of the β-sandwich domain [32]. This secretion of TGase 2 may lead to the development of renal fibrosis in tubular epithelial cells. This export mechanism is absolutely dependent on the N-terminus of TGase 2 (a.a. 88–106). Although this region contains the fibronectin-binding domain, export is independent of fibronectin binding. This region was also identified as the GK921-binding site [24]. Is this a coincidence, or does it reflect another important feature of TGase 2?

### 2.4. Triple Complex of TGase 2 in Cancer

Transglutaminase 2 also forms a triple complex within the cell. Specifically, in RCC, TGase 2 forms a triple complex with p53 and p62 [30]. In RCC, TGase 2 is highly overexpressed, and the high level of TGase 2 depletes the p53 tumor suppressor by promoting its autophagic degradation [18,25,30,43,59]. p53 is mutated in less than 4% of total RCC cases [30]. Consequently, TGase 2 knockdown or inhibition in RCC cells stabilizes p53, inducing apoptosis [18,25,30,43,59]. The mechanism of p53 depletion by TGase 2 involves the p53–TGase 2–p62 triple complex, which is degraded in the autophagosome [30]. A binding assay using a series of deletion mutants of p62, p53, and TGase 2 revealed that the N-terminus of p62 (a.a. 85–110) directly interacts with the C-terminus of TGase 2 (a.a. 592–687) while the N-terminus of p53 (a.a. 15–26) simultaneously interacts with the N-terminus (a.a. 1–139) [30]. p53 binding itself is not sufficient to form a polymer of p53, which requires the high concentration of calcium present in the autophagosome [60]. Together, these observations suggest that TGase 2 acts as a chaperone of p53 with cross-linking activity. This p53- TGase 2–p62 interaction of triple complex has an advantage of rapid autophagy to compare to mouse double-minute 2 homolog (MDM2) mediated p53 ubiquitination, which is directly destined to microtubule-associated protein light chain 3 (LC3) on the membrane of autophagosome through p62 binding in RCC [30]. TGase 2-mediated autophagy is beneficial for cancer cells because, in addition to depleting p53, it also provides molecular building blocks for cell growth. However, because many kinds of cancer have high p53 mutation rates, TGase 2 may have differential selectivity for binding proteins in other cancer types. In glioblastoma, upregulation of TGase 2 expression by microglia-derived cytokines induces CCAAT-enhancer-binding protein β (C/EBPβ) expression by depleting DNA damage-inducible transcript 3 (GADD153), triggering mesenchymal transdifferentiation of glioma stem cells [61]. To make a cross-link between GADD153 and other binding proteins, one molecule of TGase 2 must form a complex with two other proteins, or alternatively, two TGase 2 molecules can bind each other through their C-terminal domains while one partner binds another protein with its N-terminus. Subsequently, calcium triggers TGase 2 cross-linking activity, resulting in formation of a covalent bond between the binding proteins. This implies that TGase 2 in cells forms a triple complex with interacting proteins at all times.

## 3. Discovery of Allosteric Site of TGase 2

Previously, we showed that the TGase 2 inhibitor GK921 promoted RCC by stabilizing p53 [25]. The inhibitory mechanism of GK921 involves allosteric binding at the N-terminus of TGase 2 (a.a. 81–116), which triggers a conformational change that accelerates polymerization and inactivation of the enzyme [24] (Figure 1). Transglutaminase assays revealed that GK921 inhibits TGase 2 via direct binding [29]. Although the binding region is about 160 a.a. away from the active site, GK921 can nonetheless inhibit TGase 2 activity. Only the conformation change can be expected by the binding of GK921 at N-terminus as an allosteric binding site of TGase 2. This conformation change may also be linked to the acceleration of TGase 2 polymerization via noncovalent interactions at high temperature [29]. Due to the low solubility of GK921, its binding pocket has not been identified through crystallography. However, to confirm that the binding site of GK921 is in the N-terminus, a quadruple point mutant targeting charged amino acids in the N-terminus (Q95A, Q96A, Q103A, R116A) was tested [24] (Figure 1). Specifically, HEK293 cells were treated with GK921 after co-transfection with p53 and TGase 2 wild type or quadruple mutant [24]. In the presence of 1 μM GK921, wild-type TGase 2 exhibited a 50% reduction in p53 binding, whereas the mutant exhibited no reduction [24]. This confirms that GK921 binds to the N-terminus of TGase 2 in the region containing a.a. 81–116, preventing the binding of TGase 2 to p53 [24].

The N-terminus of TGase 2 (a.a. 81–116) has another important physiological function, i.e., it overlaps with the p53-binding region [24,30]. Therefore, GK921 competes with p53 for binding at the N-terminus of TGase 2, resulting in p53 stabilization in RCC [24]. Accordingly, targeting the N-terminus of TGase 2 (a.a. 81–116) has two therapeutic benefits: inhibition of TGase 2 activity and stabilization of p53 [24]. The N-terminus of TGase 2 (a.a. 88–106) is also responsible for extracellular transportation of TGase 2 itself that is independent of fibronectin binding [32]. Independently, the N-terminus of TGase 2 (a.a. 88–106) is important for fibronectin binding [62]. Therefore, GK921 may also inhibit TGase 2 externalization or fibronectin binding, as well as enzymatic activity.

## 4. Conclusions and Future Perspectives

Our theoretical picture of the molecular mechanism by which TGase 2 can catalyze cross-linking has developed over time (Figure 2). Initially, the protein was thought to exist in a folded, globular monomer shape bound to GTP, but today it is widely accepted that the protein exists as an unfolded monomer. We assumed that TGase 2 can form an unfolded homodimer form under physiological conditions. Via non-covalent interactions, TGase 2 can efficiently form triple-complex structures with substrates or binding proteins [42]. It remains a mystery why the conformational change in the presence of calcium induces covalent cross-linking, whereas the TGase 2–substrate complex forms efficiently in the absence of cross-linking. The consensus among researchers in the field is that TGase 2 undergoes a large conformational change upon activation [40,63], implying that current efforts to target the active site based on SAR have a high risk of failing to generate therapeutic applications. Here, we have suggested that GK921 binding at the N-terminus of TGase 2 is critical for triggering both inactivation of the enzyme and p53 stabilization [24]. GK921 binding at the N-terminus of TGase 2 may also play a critical role in blocking cell adhesion by inhibiting fibronectin binding, as well as blocking extracellular transportation of TGase 2, through binding at the N-terminus. Although the detailed molecular mechanism of the conformational change remains to be elucidated, the findings described here suggest that it should be possible to develop inhibitors targeting a.a. 81–116 of TGase 2 by screening chemical libraries.

## Figures and Tables

**Figure 1 medsci-06-00087-f001:**
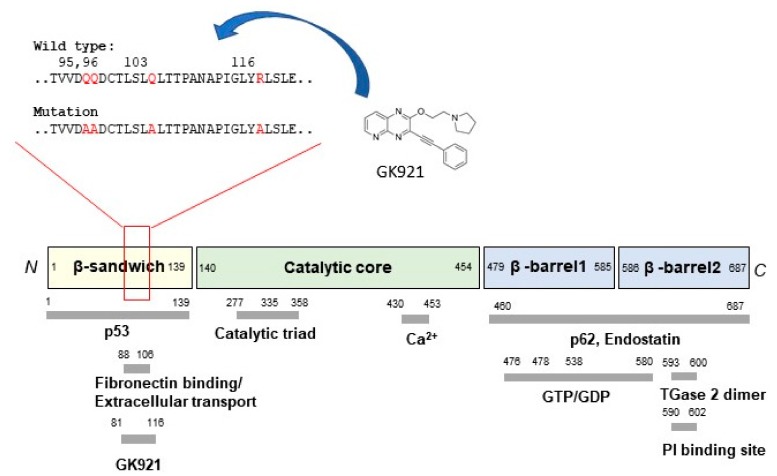
The four domains of transglutaminase 2 (TGase 2), presented with binding domains. The catalytic core domain of TGase 2 is responsible for its enzymatic activity, which is induced by Ca^2+^ (a.a. 430–453). The β-barrel 1 domain of TGase 2 contains a heparin-binding site (a.a. 262–265) [28] and a guanosine tri/diphosphate (GTP/GDP)-binding site (a.a. 476–478 and 538–580) [29,30,31]. The C-terminal β-barrel 2 domain of TGase 2 contains a heparin-binding site (a.a. 598–602) [28] and the dimerization motif (a.a. 593–600) [29]. Together, the combined β-barrel 1 and 2 domains of TGase 2 contain binding sites for p62 (a.a. 460–687) [30] and endostatin (a.a. 460–687) [31]. TGase 2 contains an extracellular trafficking sequence at the N-terminus (a.a. 88–106) of the β-sandwich domain [32]. The TGase 2 quadruple point mutant (Q95A, Q96A, Q103A, R116A) cannot bind GK921 [24].

**Figure 2 medsci-06-00087-f002:**
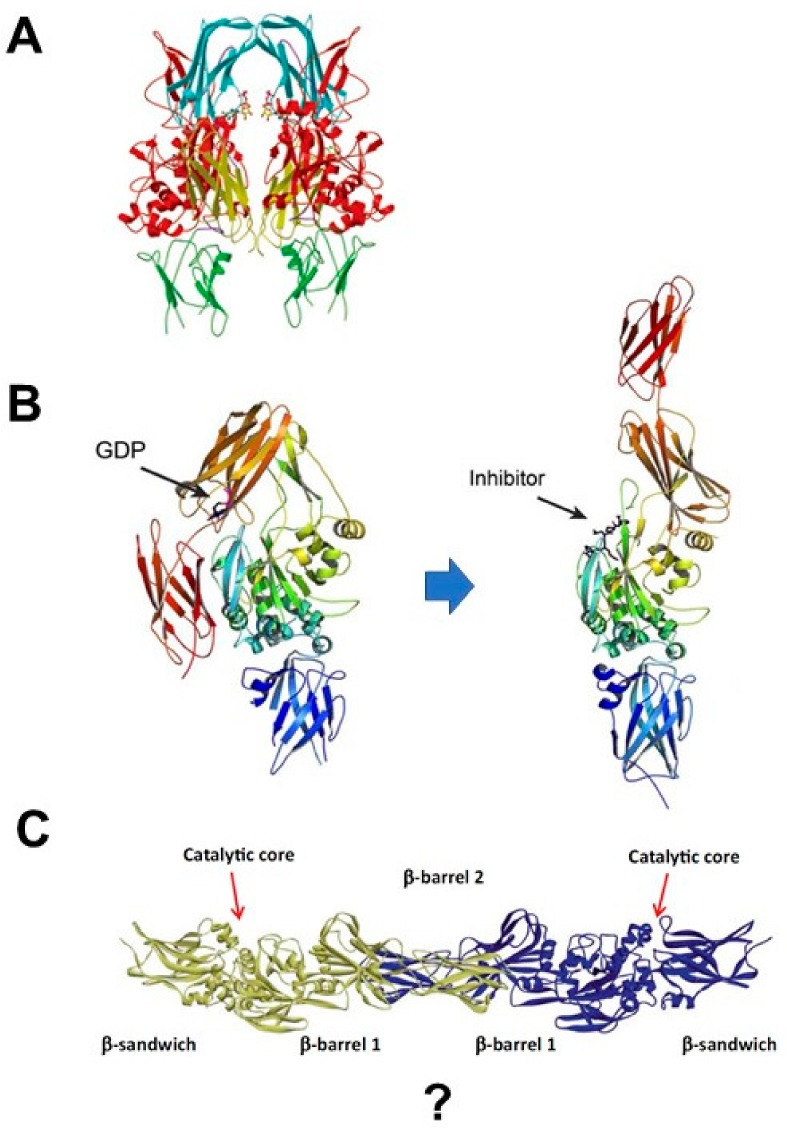
Development of the TGase 2 structure. (**A**) Human tissue TGase 2 dimer with bound GDP. Transglutaminase 2 is shown as a ribbon drawing with the β-sandwich domain, catalytic core domain, and first and second β-barrel domains shown in green, red, cyan, and yellow, respectively. The loops connecting the first β-barrel domain to the catalytic core and the second β-barrel are shown in purple. Guanosine diphosphate (GDP) is shown as a ball-and-stick model, located between the catalytic core and the first β-barrel. This figure was obtained from the original article [39] with permission from PNAS USA (Copyright (2002) National Academy of Sciences, USA). (**B**) Crystal structures of folded and unfolded TGase 2 are shown as ribbons. The N-terminal β-sandwich is shown in blue (N), the catalytic domain (Core) in green, and the C-terminal β-barrels (β1 and β2) in yellow and red, respectively. GDP-bound TGase 2 (left). Transglutaminase 2 inhibited by the active-site inhibitor Ac-P(DON)LPF-NH2 (right). This figure was obtained from the original article [40] under open-access license “CC-BY.” (**C**) Model of the unfolded dimer of TGase 2, based on the crystal structure of the open conformation (PDB:2Q3Z). Transglutaminase 2 consists of four domains: the N-terminal domain, catalytic domain (verdigris), C-terminal domain 1 (green), and C-terminal domain 2 (yellow). Reprinted by permission from Springer Nature, from [29].

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
