# Peer review of "New Insights into Development of Transglutaminase 2 Inhibitors as Pharmaceutical Lead Compounds"

_medsci, 2018, doi:10.3390/medsci6040087_

Round 1
Reviewer 1 Report
The review, authored by Soo-Youl Kim, is focused on the possibility to develop transglutaminase-2 (TG2) inhibitors able to bind the allosteric site of this enzyme. This intriguing review underlines the option of modulating the activity of TG2 modifying its protein conformation and not acting on the catalytic core.
The review is quite exhaustive and the citations reported are sufficient to cover most of the knowledge about the topic. My general opinion about this work is positive, I believe that it could be better suitable for publication on the Special Issue of Medical Sciences after some minor revisions.
Minor revisions:
1. In the introduction section, a short overview on the TG2 activities in physiological and pathological conditions should be added (as also mentioned in the Abstract), as well as a brief summary of the canonical inhibitors of TG2.
2. The quality of Figure 2 should be improve, if possible.
3. English requires polishing.
Author Response
1. In the introduction section, a short overview on the TG2 activities in physiological and pathological conditions should be added (as also mentioned in the Abstract), as well as a brief summary of the canonical inhibitors of TG2.
A1: As we have mentioned it in the abstract, we added following sentences in the beginning of Introduction.
“Prospective benefits of therapeutic approach through TGase 2 inhibition in clinical side Physiological functions and reaction mechanisms of TGase 2 have been studied for over six decades. TGase 2 is widely distributed in tissues, and it has been proposed to have various functions in context dependent fashion [1-3]. TGase 2 is responsible for the change of normal physiology including fibroblast function [4], wound healing [5], clearance of apoptotic cells [6], macrophage phagocytosis [7], glucose tolerance [8] as well as plays a key role in the development of disease pathogenesis including various cancers [9-11], several neurological disorders including Huntington’s, Alzheimer’s and Parkinson’s [12-14]. More reactions and biological functions of TGase 2 are discussed in the reviews [1,15] including NF-kB activation through I-kBa inhibition [16], HIF-1a activation through VHL inhibition [17] and suppression of apoptosis in cancer through p53 inhibition [15,18].
It seems that irreversible inhibitors are attractive for development of TGase 2 inhibition. In the field of TGase 2 inhibitor development as a prospective clinical candidate, most of research endeavors focuses on targeting active site as an irreversible inhibitor [19]. Several clinical trials are being tested for TGase 2 inhibitors. Cysteamine known as targeting active site cysteine residue has been launched for broad disease indications including cystic fibrosis [20], neuro degenerative diseases [21], Huntington’s disease [22], and so on by Raptor pharmaceuticals, Mylan, and European Institute for Cystic Fibrosis Research [19]. Zedira has focused on developing peptidomimetic TGase 2 inhibitor mimicking TGase 2 substrate for celiac disease. CHDI and Evotec develops different class of Michael acceptor compounds targeting TGase 2 active site for the treatment of diseases such as neurodegenerative diseases and celiac disease [23]. However, new finding of GK921 mechanism in TGase 2 inhibition [24] suggests that reversible inhibitors should be considered as an important position for pharmaceutical lead compound although irreversible TGase 2 inhibitors are dominant in the development pipeline.”
2. The quality of Figure 2 should be improve, if possible.
A2. I improved Fig 2 as best as I could.
3. English requires polishing.
A3. Professional English Proofing has been done (certificate is attached).

Reviewer 2 Report
The article entitled "New Insights into Development of Transglutaminase 2 Inhibitors as Pharmaceutical Lead Compounds" by Soo-Youl Kim is an updated review which discuss the possibility of developing TG2 inhibitors that target the allosteric binding site of TG2.
Minor text/language corrections are required :
pag. 2, row 11: the more updated abbreviation of Transglutaminase 2 (TG2) should be used
pag. 10, row 169 : this
pag. 12, row 218 : mdm
Author Response
pag. 2, row 11: the more updated abbreviation of Transglutaminase 2 (TG2) should be used.
A. I have added the updated abbreviation as “Transglutaminase 2 (EC 2.3.2.13, TG2 or TGase 2)”.
Folk and Cole identified an isozyme from Guinea pig liver, “TGase C” called after protein purification (C for cytosol and the -ase suffix to distinguish transglutaminase from triacylglycerol) or “TGase 2” named after gene cloning (2 from the 2nd cloned TGase in the TGase family; the product of the gene named TGM2), and characterized the TGase enzymatic reaction as an acyltransfer of peptide-bound glutamine to polyamines or peptide-bound lysine (Folk 1980; Chung and Folk 1970). I prefer to use “TGase 2” in my paper to avoid confusion with an abbreviation of “TG” that commonly represents “triglyderide” in biochemistry as well as scientific heritage after Drs. Folk and Chung.
pag. 10, row 169 : this
A. which was changed to “the”
pag. 12, row 218 : mdm
A. which was changed to “mouse double minute 2 homolog (MDM2)”